# Genetic susceptibility to angiotensin-converting enzyme-inhibitor induced angioedema: A systematic review and evaluation of methodological approaches

**Haivin Aziz Ali[1], Anne Fog Lomholt[2], Seyed Hamidreza Mahmoudpour[3], Thorbjørn Hermanrud[2], Anette Bygum[4], Christian von Buchwald[2], Marianne Antonius Jakobsen[5], Eva Rye Rasmussen[2] ***

**1** Faculty of Health and Medical Sciences, University of Copenhagen, Copenhagen, Denmark, **2** Department of Oto-Rhino-Laryngology—Head and Neck Surgery and Audiology, Denmark, **3** IMBEI—Institute of Medical Biostatistics, Epidemiology and Informatics, University Medical Center of the Johannes Gutenberg, CTH -Center for Thrombosis and Hemostasis Mainz, Mainz, Germany, **4** Department of Dermatology I and Allergy Center, Odense University Hospital, Indgang, Odense C, Denmark, **5** Department of Clinical Immunology, Odense University Hospital, Denmark, Odense C, Denmark

* eva.rye.rasmussen@dadlnet.dk

## Abstract

Angiotensin-converting enzyme (ACE) converts angiotensin I to angiotensin II which causes vasoconstriction. ACE inhibitors reduce blood pressure by inhibiting ACE. A well-known adverse drug reaction to ACE inhibitors is ACE inhibitor-induced angioedema (ACEi-AE). Angioedema is a swelling of skin and mucosa, which can be fatal if the airway is compromised. We have performed a systematic review of the evidence suggesting that genetic polymorphisms are associated with ACEi-AE and evaluated the methodological approaches of the included studies. The Cochrane Database of Systematic Reviews, Google Scholar, and PubMed were searched. Studies investigating the association between genetic markers and ACEi-AE were included. The Q-genie tool was used to evaluate the quality of the study methodologies. Seven studies were included. With the exception of one whole genome study, all of the included studies were candidate gene association studies. Study quality assessment scores ranged from 36 to 55. One study was found to be of good quality, suggesting that the detected associations may be unreliable. The inferior quality of some studies was due to poor organization, lack of analyses and missing information. Polymorphisms within *XPEPNP2*, *BDKRB2*–9/+ 9 and *neprilysin* genes, were reported to be associated with increased risk of ACEi-AE. However, due to low quality, these associations need to be confirmed in larger studies.

## Introduction

Angiotensin-converting enzyme inhibitors (ACEis) are used to treat cardiovascular diseases and diabetic nephropathy [1]. A 3-year follow-up of cardiovascular disease patients found that

**Data Availability Statement:** All relevant data are within the paper and its Supporting Information files.

**Funding:** The authors received no specific funding for this work.

**Competing interests:** The authors have declared that no competing interests exist.

ACEi treatment reduced mortality by 18% compared to a placebo-treated group [2]. ACEis inhibit the conversion of angiotensin I to angiotensin II by angiotensin-converting enzyme (ACE). Angiotensin II is a vasoconstrictor which increases blood pressure [3]. ACEis are prescribed to approximately 40 million people worldwide. Up to 0.7% of ACEi users develop angioedema which is a well-known adverse drug reaction (ADR) [4]. Angioedema is a non-itchy swelling of the deeper layers of the skin or mucosa and may occur in various parts of the body. This symptom may be fatal when it leads to airway obstruction [5]. Upper airway swellings have occured in as many as fifty percent of ACEi-induced angioedema (ACEi-AE) cases [6]. Angioedema can develop within a few minutes, blocking the respiratory tract. This requires emergency treatment including intubation or in very severe cases, acute tracheotomy[7].

The pathophysiology of ACEi-AE has been linked to bradykinin (BK) and potentially substance P [8,9]. BK is a potent vasodilator which enhances the permeability of capillaries leading to fluid leakage into the surrounding tissues [10]. In healthy individuals, BK is degraded by ACE, aminopeptidase P (APP), neutral endopeptidase, dipeptidylpeptidase IV and carboxypeptidase N [11–13]. A small amount of BK is converted into the active metabolite, des-arginine-9-bradykinin, by the enzyme carboxypeptidase N. This metabolite is also degraded by ACE and APP [12].

The risk of developing ACEi-AE may be affected by a patient's ethnicity or sex, and also by smoking, seasonal allergies or immunosuppressant treatment [14]. Patients who are black and/or female have an increased risk of developing ACEi-AE, whereas diabetes seems to be protective [14,15]. Several studies have investigated associations between genetic variations and ACEi-AE but the results have been inconsistent [16–22]. This study systematically reviews the current literature on genetic susceptibility to ACEi-AE and evaluates the methodology of included studies using the newly developed "quality of genetic associations studies tool" (Q-genie) [23].

## Materials and methods

### Protocol and search strategy

The search period was set up until August 2018. A complementary literature search was performed in September 2019 and no further relevant papers were published since the time of our original search. The Cochrane Database of Systematic Reviews, Google Scholar, and PubMed were searched to retrieve the relevant papers based on the predefined search terms and strategy. To maximize the search coverage, a combination of medical subject heading (MeSH) terms for "angiotensin converting enzyme inhibitors", "angioedema", "genetic markers", "polymorphisms", and "pharmacogenetics" were included. Furthermore the bibliographies of retrieved papers were assessed to identify further studies.

The Preferred Reporting Items for Systematic Reviews and Meta-Analyses (PRISMA) guidelines were followed for this systematic review and our protocol was registered and published with the Prospective Register of Systematic Reviews (PROSPERO) ref. CRD42016041639 [24,25].

### Eligibility criteria

Studies were eligible if they included patients who developed angioedema during treatment with ACEis. Observational case-control studies, clinical trials, cohort studies, case reports, case series, and other observational or experimental studies were all considered eligible. Reviews, studies not written in English, animal studies, conference abstracts (if the full study was not published) and studies of angioedema induced by other conditions or medication were all excluded [25].

### Selection of studies and data extraction

After removing duplicates, the study titles and abstracts were screened independently by two reviewers (ERR, TH) to exclude those that did not meet the inclusion criteria. The full-text of each remaining study was evaluated. Disagreements between the reviewers were resolved by discussion. Studies were included if they investigated the relationship between genetic polymorphisms and ACEi-AE. The following information was extracted for each included study: first author's name, publication year, study design and main findings of the studies. Two independent reviewers checked the information for precision (ERR, TH) [25].

### Quality assessment

Four reviewers (HAA, ERR, AFL, SHM) independently assessed the quality of each included study using Q-genie tool. This tool is specifically designed to facilitate quality assessment of genetic studies and to evaluate the risk of bias [23]. The Q-genie tool consists of 11 questions which address the following aspects of study methodologies: study rationale, outcome, comparability, exposure, bias, sample size, analyses, statistical methods and control for confounding, inferences for genetic analyses and inferences drawn from results.

Each question was scored from 1–7 as follows: "1 (poor)", "2", "3 (good)", "4", "5 (very good)", "6" or "7 (excellent)". For studies with control groups, a total score of ≤35 indicates poor quality, a score of 36–45 indicates moderate quality and a score of >45 indicates good quality. In studies without control groups, a total score of ≤32 indicates poor quality, a score between 33–40 indicates moderate quality and a score of > 40 indicates good quality [23].

### Ethics

This study did not require any approval from the ethics committee.

## Results

### Study selection

The strategy described in materials and methods was used to search electronic bibliographic databases (Cochrane Database of Systematic Reviews, Google Scholar, and PubMed) and 404 studies were identified. Next, 261 duplicates were removed, and the remaining 143 studies were screened for content. Of these studies, 119 did not meet the inclusion criteria. The full-text of each of the remaining 24 studies was assessed, and 17 studies were excluded. The detailed study selection procedure is shown in Fig 1. Seven studies met all the inclusion criteria.

### Quality assessment

The seven included studies were scored using the Q-Genie tool [16–22]. Table 1 provides an overview of the included studies, their characteristics and the total scores for each of the selected studies, using the Q-genie tool. S1 Table provides detailed information on the scores of each study. One study by Paré et al. was of "good quality" and the remaining six studies were graded of "moderate quality".

### Study characteristics

The included studies were published between 2005 and 2013. All seven studies had a case-control study design. The total number of included patients with ACE-AE was 547. Five studies described the ethnicity of patients. Three study populations were of mixed ethnicity; two of

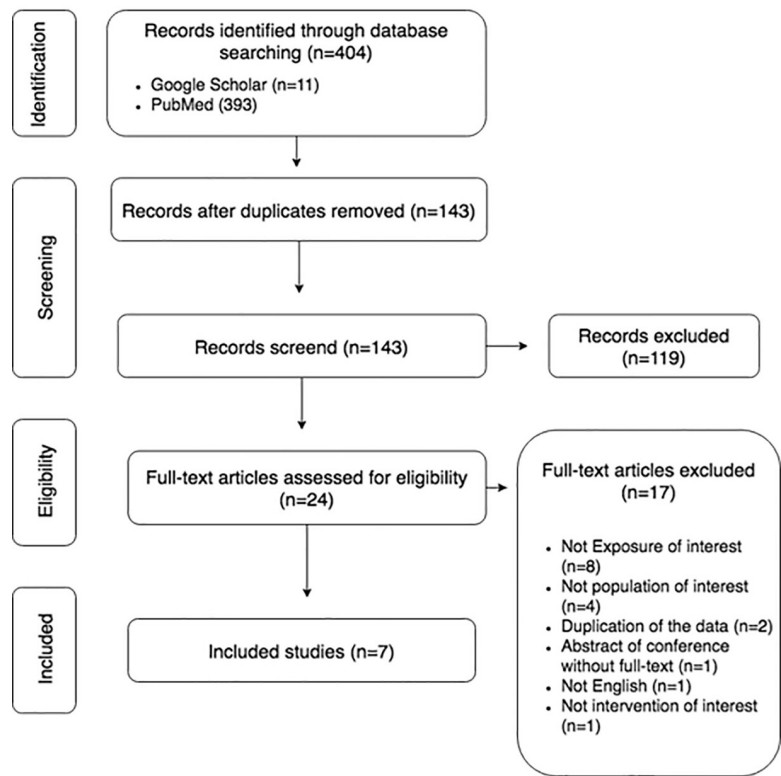

PRISMA: Preferred Reporting Items for Systematic Reviews and Meta-Analyses

**Fig 1. PRISMA flow diagram showing study selection procedure.**

these studies included black and Caucasian patients, and the third study population consisted of blacks, Caucasian and Cape mixed-race patients. The remaining two studies had only included Caucasians patients[17,22].

Three of the studies examined the relationship between the *ACE* I/D polymorphism and ACEi-AE, and none of them found any statistically significant associations [16,17,21]. In two of these studies, the relationship between the *Bradykinin B2 receptor* gene and ACEi-AE was also investigated, and four *Bradykinin receptor B2* gene polymorphisms were evaluated. There was reported a significant association between the *Bradykinin receptor B2*–9 /+ 9 and the risk of ACEi-AE [16,21]. The other studied polymorphisms (*BDKRB2* C-58T, *BDKRB2* 2/3 *and BDKRB2* c.C181T) showed no significant associations with ACEi-AE.

Three of the studies suggested that ACEi-AE development was due to a lower concentration of APP. The *XPNPEP2* C-2399A genotype was associated with ACEi-AE [18]. In one of the studies, an ATG haplotype in the *XPNPEP2* 5'-regulatory region was associated with a greater variance in plasma APP and thus increased the risk of developing ACEi-AE, due to a decreased degradation of BK [20].

A genome-wide association study (GWAS) was conducted in only one of the included studies [19]. No statistically significant correlation was found between any single nucleotide polymorphisms (SNP) and ACEi-AE after the Bonferroni correction for multiple testing was applied. However, moderate evidence for an association with ACEi-AE was reported ($p < 10^{-4}$) for 16 SNP's in black patients and for 41 SNPs in Caucasian patients [19]. In this study, a candidate gene analysis was additionally performed to investigate polymorphisms in the following

**Table 1. Outcomes reported by studies investigating the relationship between ACEi-induced angioedema and gene polymorphisms.**

| First author and year of publication | Study design | Country | Ethnicity | Age means ± SD (years) | Gender (F, %) | Participants (treatment duration with ACEi) | Genotyping | Polymorphisms | Main findings | The total score based on the Q-genie tool |
|---|---|---|---|---|---|---|---|---|---|---|
| **Bas 2009** [16] | Case-control study | Germany | NS | 62 ± 1,6 | 47 | 65 cases (36,7 ± 4,8 months) 65 controls (49 ± 3,5 months) | Allele-specific PCR | *ACE* I/D (rs 4646994) *BDKRB2* 2/3 *BDKRB2* c. C181T (*BDKRB2* c.40>T rs1046248) | No significant association was found between *ACE* I/D or *BDKRB2* 2/3 and c. C181T (*BDKRB2* c.40>T rs1046248) gene polymorphism and ACEi-AE | 42 |
| **Duan 2005** [22] | Case-control study | USA, Canada, Belgium | Caucasian | NS | NS | 20 cases (less than 8 years) 60 controls | Allele-specific PCR | *XPNPEP2* C-2399A (*XPNPEP2* c.-2400C>A rs3788853) | *XPNPEP2* C-2399A (*XPNPEP2* c.-2400C>A rs3788853) is correlated with decreased APP activity and thus a greater incidence of ACEi-AE | 39 |
| **Gulec 2008** [17] | Case-control study | Turkey | Caucasian | 58,06 ± 8,71 | 72,4 | 32 cases (less than 36 months) 46 controls (NS) | PCR | *ACE* I/D | No association was found between *ACE* gene polymorphism and ACEi-AE or ATRB-AE | 36 |
| **La Corte 2011** [20] | Case-control study | USA, Canada, Belgium | NS | NS | NS | 34 cases (NS) 127 controls (NS) | TaqMan SNP genotyping | *XPNPEP2* C-2399A (*XPNPEP2* c. C.-2400C>A rs3788853) | A functional ATG haplotype was found in the 5'regulatory region of *XPNPEP2*; this region is correlated with an increased risk of ACEi-AE due to decreased APP activity. The ATG haplotype is more informative than the *XPNPEP2* C-2399A polymorphism (*XPNPEP2* c. C.-2400C>A rs3788853) | 37 |
| **Moholisa 2013** [21] | Case-control study | South Africa | Black, Caucasian and Cape mixed ancestry | 49 | 79 | 52 cases (NS) 77 controls (2 years) | Allel-specific PCR and RFLP | *ACE* I/D *BDKRB2*–9/+9 *BDKRB2* C-58T (*BDKRB2* c.-192T>C rs1799722) | The association between Bradykinin receptor B2–9 /+ 9 and the development of ACEi-AE and ACE-cough was significant. No significant correlation shown between *ACE* I/D or *BDKRB2* C-58T (*BDKRB2* c.-192T>C rs1799722) polymorphism and ACEi-AE and ACE-cough | 40 |

*(Continued)*

**Table 1.** (Continued)

| First author and year of publication | Study design | Country | Ethnicity | Age means ± SD (years) | Gender (F, %) | Participants (treatment duration with ACEi) | Genotyping | Polymorphisms | Main findings | The total score based on the Q-genie tool |
|---|---|---|---|---|---|---|---|---|---|---|
| **Pare**[19] | Case-control study | USA | Black and Caucasian | 58,4 ± 14,1 | 54,9 | 175 cases 489 controls *(at least 6 months)* | GWAS | *CPN MME XPNPEP2 DPP4 BDKRB1 BDKRB2 TACR1* | A GWAS study investigated the relationship between the SNPs and ACEi-AE; no genome-wide significant connection was found. However, there was moderate evidence that 16 SNPs from African-Americans and 41 SNPs from European-Americans wew associated with ACEi-AE (p $<10^{-4}$). In a candidate gene analyses, there was a significant correlation between *MME* and ACEi-AE | 55 |
| **Woodard-Grice 2010** [18] | Case-control study | USA | Black and Caucasian | 57,3 ± 14,1 | 56,8 | 169 cases *(median of 5 months)* 397 controls *(median of 42 months)* | Allele-specific PCR | *XPNPEP2* C-2399A (*XPNPEP2* c.C-2400C>A rs3788853) | The *XPNPEP2* C-2399A (*XPNPEP2* c.C-2400C>A rs3788853) genotype was correlated with a greater risk of ACEi-AE in men | 45 |

ACE: angiotension-converting enzyme; ACEi: ACE-inhibitor; ACEi-AE: ACE-inhibitor induced angioedema; ATRB-AE: Angiotensin II receptor blocker induced angioedema; *ACE* I/D: *ACE* insertion/deletion; APP: aminopeptidase P; ATRB: angiotensin receptor blocker; F: female; NS: not specified; OR: odds ratio; BDKRB2: bradykinin B2; BDKRB1: bradykinin B1; CPN: carboxypeptidase P; MME: neprilysin; XPNPEP2: aminopeptidase P; TACR1: NK1 receptor; GWAS: genome-wide association study; SD: standard deviation; Q-genie tool: quality of genetic association studies; RFLP: restriction fragment length polymorphism; PCR: polymerase chain reaction.

genes: *bradykinin B2*, *bradykinin B1*, *carboxypeptidase P*, *neprilysin*, *APP* and the *NK1 receptor*. One polymorphism in the *neprilysin* gene was associated with ACEi-AE [19].

## Discussion

This is the first systematic review to investigate the relationship between genetic markers and ACEi-AE using the Q-genie tool to evaluate the methodological approaches. The included studies examined different polymorphisms of *ACE* I/D polymorphism, *XPNPEP2* and the *Bradykinin B2 receptor* genes. In one of the studies, a GWAS and a subsequent candidate gene analysis were performed.

The *ACE* gene has a functionally relevant polymorphism. There is an alu repetition that is either present (I-allele) or absent (D-allele). It is suggested that the *ACE* D-allele is associated with higher serum enzyme activity than the I-allele [16]. This could mean that when the D allele is present, bradykinin is more readily degraded. The polymorphism of *ACE* I/D was not

significantly associated with ACEi-AE. However all the three studies investigating this polymorphism were underpowered, and this could have affected the findings [16,17,21].

Since it has been shown that a low plasma activity of APP could be a predisposing factor for the development of angioedema during ACEi treatment [26], in three studies the polymorphism *XPNPEP2* of the *APP* gene was investigated, and a significant association with ACEi-AE was identified consistently in all three studies [18,20,22]. Those three studies included a total of 223 cases and 584 matched controls. However, there appeared to be some overlap between the study populations by Duan et al. and Cilia La Corte et al.[20,22]. A functional haplotype was found in the 5'-regulatory region of *XPNPEP2*. This correlated with an increased risk of ACEi-AE due to decreased APP activity [20]. This haplotype could therefore be a marker for an increased risk of developing ACEi-AE, since decreased APP activity would potentially cause delayed BK degradation[20]

The GWAS investigated the relationship between the SNPs identified and ACEi-AE but no significant associations were found [19]. A total of 57 SNPs were identified, but there was only moderate evidence that any of these were associated with ACEi-AE. In addition, this moderate evidence was inconclusive because the associated *p*-values did not reach the level usually required in GWAS studies ($p < 10^{-8}$). The authors concluded that they were unable to establish any significant genome-wide association and suggested that the development of ACEi-AE may be influenced by multiple genetic factors. A candidate gene analysis was performed in spite of the non-significance of the GWAS findings, and a significant association was found between polymorphisms of the *neprilysin* gene and ACEi-AE [19].

In the study by Moholisa et al. they found a significant association between the genotype $B_2$ + 9 / -9 and ACEi-AE, this significance was not found for the genotype $B_2$−9 / -9. It should be noted that the control group was not in Hardy-Weinberg equilibrium, which affects the credibility of the association negatively [21]. Moreover, Bas et al. found no significant association between this polymorphism and ACEi-AE.

All of the included studies were evaluated using the Q-genie tool and achieved "moderate " or "good" quality scores. The scores are presented in Table 1 and Fig 2. Table 1 shows that the study by Pare et al. received the highest score (55 points) and was of good quality [19]. This study was carefully designed and described in detail. Fig 2 shows this study's scores in each of the 11 aspects; its lowest score (3) was the "*Non-technical classification of the exposure*" (score 3).

The study that scored the lowest was by Gülec et al., shown in Table 1 and Fig 2 [17]. It received a score of 36 and was of moderate quality. This study was not particularly informative, and it failed to describe the procedures and how the results were achieved in sufficient detail.

Fig 2 shows the overall scores for each study and the scores for each of the 11 questions obtained by each study. No study scored seven (excellent) for any of the questions. For question five: "*Non-technical classification of the exposure*" one study scored four and all the remaining studies scored less [17]. This indicates that the studies had not adequately reported how they performed the non-technical classification of the genetic variants, meaning that it is not clear whether it was a blinded assessor performed the genotyping. For question seven: "*Sample size*", two studies scored four the remaining studies scored less due to small sample sizes and the lack of a power calculation[18,19]. Obtaining a sufficient sample size, and thus sufficient power, is difficult because ACE-AE is a rare adverse drug reaction. None of the studies performed a power calculation. In this regard, a collaboration between the different expert groups on ACEi-AE could minimize the issue of power by data sharing. One such group has already been formed in the PREDICTION-ADR consortium recruiting cases from five different countries, but further collaboration would be beneficial [27].

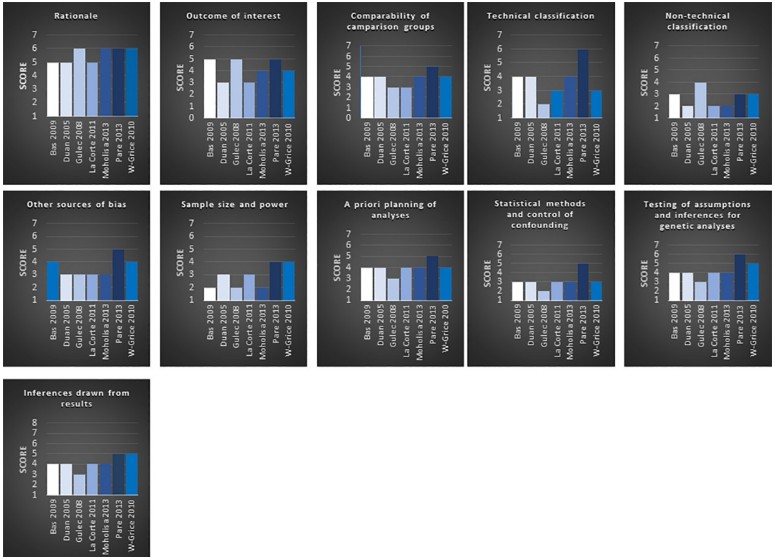

**Fig 2. Scores for each study and the distribution of the 11 methodological aspects.**

Question nine, "*Statistical methods and control for confounding*" evaluated the methods used to control for confounding variables, ensures that missing sample and genetic variant data have been correctly handled and checks that each study has tested for false positive results. Only one study scored five, and the remaining scored three or less [19]. The lower scores indicate that these concerns had not been adequately discussed. This is noteworthy because failing to consider the possible effect of confounding variables can lead to incorrect conclusions.

Many aspects need to be considered when designing a study, and it is important to describe procedures in detail to ensure the research can be replicated for further validation.

A recent case control study by Hubers et al. found that bradykinin level in acute ACEi-AE was elevated compared to treated controls, but there was no evidence of increased production through high-molecular-weight kininogen [28]. Also levels of the bradykinin degradation product bradykinin 1–5 (via ACE and neprilysin) was not increased even though the total levels were increased, which suggest impairment of non-ACE degradation pathways i.e. via neprilysin. This might suggest that polymorphisms in genes regulating these pathways would be associated with ACEi-AE; however Hubers et al. did not discuss the fact that increased levels of bradykinin can also arise due to increased production as is seen in some forms of hereditary angioedema [29].

Another issue regarding ACEi-AE is that some cases might suffer from rare types of angioedema i.e. hereditary angioedema associated with factor XII mutations [30,31], which could be 'set off' by the ACE inhibition. In the future it would be desirable to genotype all ACEi-AE patients at least for the most common forms of hereditary angioedema.

Among the included studies, there were some differences in nomenclature used to describe polymorphisms. The Human Genome Variation Society (HGVS) nomenclature is recommended for clinical diagnostic reporting. This system includes a formats for describing variants with reference to sequence information, which can unambiguously specify positional information [32]. To avoid misunderstandings this nomenclature should be adhered to in future studies.

The Q-genie tool has some limitations. Using Q-genie to review a large number of studies is time consuming: it takes approximately 10 hours to review 30 studies, however it is

estimated that Q-genie is more efficient than other tools, and reviewers will become more proficient at using Q-genie with practice [23]. Another limitation is that a study can have low scores for some of the 11 questions and still be "good" quality. In addition, despite the results, a good quality study can have a low score and therefore be considered low quality due to insufficient reporting of methods [33].

## Conclusions

This is the first systematic review that used the Q-genie tool to evaluate the quality of studies investigating ACEi-AE. Only one study was evaluated as "good quality". None of the studies performed a power calculation which is important for assessing the number of observations needed to detect a statistically significant association. The inferior quality of some studies was due to poor organization, lack of analyses and missing information. We recommended that these aspects are considered before undertaking future genetic studies because study reproducibility is very important. A collaboration between the different expert groups on ACEi-AE could minimize the issue of power by data sharing.

The polymorphisms in *XPEPNP2*, *BDKRB2*–9/+ 9 and *neprilysin* genes, were associated with an increased risk of developing ACEi-AE. However, due to low quality, these associations need to be confirmed in larger studies.

## Supporting information

**S1 Table. Details on individual scores for the included studies based on Q-genie tool.** (DOCX)

## Author Contributions

**Conceptualization:** Anette Bygum, Christian von Buchwald, Marianne Antonius Jakobsen, Eva Rye Rasmussen.

**Data curation:** Haivin Aziz Ali, Seyed Hamidreza Mahmoudpour, Thorbjørn Hermanrud, Marianne Antonius Jakobsen, Eva Rye Rasmussen.

**Formal analysis:** Haivin Aziz Ali, Anne Fog Lomholt, Seyed Hamidreza Mahmoudpour, Thorbjørn Hermanrud.

**Investigation:** Haivin Aziz Ali, Anne Fog Lomholt, Seyed Hamidreza Mahmoudpour, Thorbjørn Hermanrud, Marianne Antonius Jakobsen, Eva Rye Rasmussen.

**Methodology:** Anette Bygum, Christian von Buchwald, Eva Rye Rasmussen.

**Project administration:** Eva Rye Rasmussen.

**Software:** Thorbjørn Hermanrud.

**Supervision:** Anne Fog Lomholt, Anette Bygum, Christian von Buchwald, Marianne Antonius Jakobsen, Eva Rye Rasmussen.

**Validation:** Haivin Aziz Ali, Anne Fog Lomholt, Seyed Hamidreza Mahmoudpour, Anette Bygum, Eva Rye Rasmussen.

**Writing – original draft:** Haivin Aziz Ali.

**Writing – review & editing:** Anne Fog Lomholt, Seyed Hamidreza Mahmoudpour, Thorbjørn Hermanrud, Anette Bygum, Christian von Buchwald, Marianne Antonius Jakobsen, Eva Rye Rasmussen.

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
