## [Decision Letter · Decision Letter 0]

2 Sep 2019

PONE-D-19-22350

Genetic susceptibility to angiotensin-converting enzyme-inhibitor induced angioedema: a systematic review and evaluation of methodological approaches

PLOS ONE

Dear Dr. Rasmussen,

Thank you for submitting your manuscript to PLOS ONE. After careful consideration, we feel that it has merit but does not fully meet PLOS ONE’s publication criteria as it currently stands. Therefore, we invite you to submit a revised version of the manuscript that addresses the points raised during the review process.

We would appreciate receiving your revised manuscript by Oct 17 2019 11:59PM. To enhance the reproducibility of your results, we recommend that if applicable you deposit your laboratory protocols in protocols.io, where a protocol can be assigned its own identifier (DOI) such that it can be cited independently in the future. For instructions see: http://journals.plos.org/plosone/s/submission-guidelines#loc-laboratory-protocols

We look forward to receiving your revised manuscript.

Kind regards,

Michael Bader

Academic Editor

PLOS ONE

Journal Requirements:

2. We note that the original search was performed from January 2018 to August 2018. Please discuss whether relevant literature has been published in the interim that would be expected to affect the results of the meta-analysis. Additionally, please include the description of their search strategy and search terms in the methods section of the manuscript.

Reviewers' comments:

Reviewer's Responses to Questions

**Comments to the Author**

1. Is the manuscript technically sound, and do the data support the conclusions?

Reviewer #1: Partly

Reviewer #2: Yes

2. Has the statistical analysis been performed appropriately and rigorously? 

Reviewer #1: N/A

Reviewer #2: Yes

3. Have the authors made all data underlying the findings in their manuscript fully available?

Reviewer #1: No

Reviewer #2: Yes

4. Is the manuscript presented in an intelligible fashion and written in standard English?

Reviewer #1: No

Reviewer #2: Yes

5. Review Comments to the Author

Reviewer #1: Ali et al. performed a systematic review to evaluate the current evidence of an association of genetic polymorphisms with ACEi-angioedema. Among the seven studies included, significant associations were reported in four studies regarding XPEPNP2, BDKRB2 -9/+ 9 and Neprilysin gene polymorphisms. Of note, only the significant association with XPEPNP2 polymorphisms was consistently reported in three different studies. The authors conclude that the significant associations of genetic polymorphisms with ACEi-angioedema exist despite the limited quality of the studies estimated with the Q-genie tool.

Major Comments

1) The results of this systematic review reveal only minor, if any, additional information.

2) The manuscript appears to be poorly prepared. For instance, the in-text citation numbers do not match the citations of the bibliography. Examples are

line 76, citation 19,

page 7, the citation Duan 2005 in Table 1 and this citation is missing in the bibliography,

page 11, line 17, citation 15,

page 12, line 32, citation 21 is missing in the bibliography

page 13, line 56, citations 12 and 15

page 13, line 62, citation 22 is missing in the bibliography

Furthermore, some Results are presented twice as is the case for the overall quality scores which are given in Table 1 and additionally illustrated in Fig. 2. Likewise, S2 Table is superfluous as this information is included in S3 Table. The scientific message of Fig. 3 appears unclear to me as well.

The last chapter of the Abstract contains a duplication in line 36. Similarly, the same statement is given in line 123 and on page 11, line 12.

3) The Discussion is for the most part a repetition of Results. I think it would have been important to discuss the at least the APP gene polymorphisms in more detail including the description and citation of the first report of reduced APP activity in patients with ACEi-angioedema (pubmed/12086766). On the other hand, the part of the discussion about the results of Hubers et al. is not correct as an increased concentration of bradykinin isn’t necessarily caused by an impairment of bradykinin degradation (please refer to HAE). These authors based their conclusion on two further results, i.e. no change of the concentrations of the bradykinin degradation fragment B1-5 and the HMWK degradation product generated by kallikrein.

Reviewer #2: This article represents the first systematic review on genetic markers in ACEi-AE and further discuss the methodology used. It properly addresses relevant points in the evaluation of the selected articles, the weakness of each studies and limitations imposed.

In the abstract, I expected to see a summary of the reliable genetic markers associated to ACEi-AE. But the only conclusion made was about the quality of the studies. I suggest including some relevant data regarding genetic susceptibility to ACEi-AE, as implied in the title and stated in the final conclusions.

Anecdotal evidences suggest that patients which present angioedema months after the withdrawn of ACEi cannot be classified as ACEi-AE but belong to another angioedema type. Although rare, a few reports support this hypothesis, describing FXII-HAE individuals that have been asymptomatic during their whole lives and only presented angioedema at advanced age when taking ACEis.

The use of Q-genie is interesting and strength the reliability of the selected articles. My only concern is that the focus of the article seems to be the methodology used and not the results of the analyzed studies. I expected a deeper discussion on the polymorphisms studied and the positive association found between them and ACEi-AE.

Although Moholisa`s study find a significant association of +9/-9 genotype with both ACEi-cough and ACEi-AE, this association was not significant for the genotype -9/-9. Another point not highlighted is the fact that the control group was not in Hardy-Weinberg Equilibrium. The fact that Bas et al (2010) study did not found a significative association between this polymorphism and ACEi-AE also weakens its hypothetical influence in ACEi-AE. Another point to emphasize is the fact that –9/+9 was also nover significantly associated with worst prognosis in HAE, where bradykinin role is quite clear. Should you surely conclude the -9/+9 polymorphism to be associated to ACEi-AE?

I disagree that the word-count limit in scientific journal represents a limitation in properly reporting methods, as mentioned in lines 117-118. Currently, even articles published as short communications or brie reports have almost unlimited space in supplementary material.

As well pointed out by the authors, ACEi-AE in quite rare, hampering the power of all the analyzed studies. Besides the methodological recommendations for better organization to improve the quality of studies, the creation of consortium among expert groups on ACEi-AE could represent a big step into a larger and conclusive study on ACEi-AE susceptibility.

Minor

Line 43: Replace “ACEis inhibits the…” by ACEi inhibit the…”.

Line 22: The reference 3, shown in the end of the sentence, should be withdrawn. Bas et al (2010) didn`t found association between -9/+9 polymorphism and ACEi-AE.

In the Discussion (line 46), the authors say “The polymorphism of ACE I/D were significantly associated with AEi-AE. I think you wanted to say, “were not significantly associated”, right?

6. PLOS authors have the option to publish the peer review history of their article (what does this mean?). If published, this will include your full peer review and any attached files.

Reviewer #1: No

Reviewer #2: No

---

## [Author Response · Author response to Decision Letter 0]

16 Oct 2019

To the esteemed editor and reviewers

The authors thank you for your decision letter from PLOS ONE. We greatly appreciate the constructive comments. We have now addressed these comments, and believe this has strengthened the paper. On the following pages, we outline point-by-point responses to the comments by the reviewers. We look forward to hearing from you in the near future. 

• We note that the original search was performed from January 2018 to August 2018. Please discuss whether relevant literature has been published in the interim that would be expected to affect the results of the meta-analysis. Additionally, please include the description of their search strategy and search terms in the methods section of the manuscript.

Response: A follow-up literature search was performed September 2019 and no further relevant papers were published since the original search. A description of our search strategy is now included in the methods section. Table S1 has been removed since the content is now included in the manuscript.

Reviewer's Responses to Questions

• Is the manuscript technically sound, and do the data support the conclusions?

Reviewer #1: Partly

Reviewer #2: Yes

Response: The authors thank the reviewers.

• Has the statistical analysis been performed appropriately and rigorously? 

Reviewer #1: N/A

Reviewer #2: Yes

Response: The authors thank the reviewers for this notion.

• Have the authors made all data underlying the findings in their manuscript fully available? We requires authors to make all data underlying the findings described in their manuscript fully available without restriction, with rare exception (please refer to the Data Availability Statement in the manuscript PDF file). The data should be provided as part of the manuscript or its supporting information, or deposited to a public repository. For example, in addition to summary statistics, the data points behind means, medians and variance measures should be available. If there are restrictions on publicly sharing data—e.g. participant privacy or use of data from a third party—those must be specified.

Reviewer #1: No

Reviewer #2: Yes

Response: The authors have made all data available in the tables, figures and supplementary material. There are no more data than that. We will be glad to share anything, so please specify any lacking information.

• Is the manuscript presented in an intelligible fashion and written in standard English?

Reviewer #1: No

Reviewer #2: Yes

Response: The manuscript has undergone professional English language revision by Dr. Bobby Brown, Medical Writing Services Member of the European Medical Writers Association, UK. If of any interest, we can send the esteemed editor and reviewers the invoice as proof hereof. 

Review Comments to the Author

• Reviewer #1: Ali et al. performed a systematic review to evaluate the current evidence of an association of genetic polymorphisms with ACEi-angioedema. Among the seven studies included, significant associations were reported in four studies regarding XPEPNP2, BDKRB2 -9/+ 9 and Neprilysin gene polymorphisms. Of note, only the significant association with XPEPNP2 polymorphisms was consistently reported in three different studies. The authors conclude that the significant associations of genetic polymorphisms with ACEi-angioedema exist despite the limited quality of the studies estimated with the Q-genie tool.

Major Comments

1) The results of this systematic review reveal only minor, if any, additional information.

Response: The authors agree, we have now adapted the manuscript accordingly. The conclusion has been re-written as we cannot assess whether there is actually a significant association due to the low quality of the studies.

• 2) The manuscript appears to be poorly prepared. For instance, the in-text citation numbers do not match the citations of the bibliography. Examples are

line 76, citation 19,

page 7, the citation Duan 2005 in Table 1 and this citation is missing in the bibliography,

page 11, line 17, citation 15,

page 12, line 32, citation 21 is missing in the bibliography

page 13, line 56, citations 12 and 15

page 13, line 62, citation 22 is missing in the bibliography

Response: Thanks to the reviewer for mentioning this mismatch between the text and bibliography. This was due to an error in our reference-managing program which happened probably shortly before the submission. The authors have been through the citations and adapted the manuscript.

• Furthermore, some Results are presented twice as is the case for the overall quality scores which are given in Table 1 and additionally illustrated in Fig. 2. Likewise, S2 Table is superfluous as this information is included in S3 Table. The scientific message of Fig. 3 appears unclear to me as well.

Response: In accordance to the reviewer’s comment, we have now adapted the manuscript. There is a consensus that Fig. 2 and S2 Table are removed. However, we believe that Fig. 3 (now figure 2 in the manuscript) illustrates nicely, where the problem lies precisely with the quality of each study. In addition, the authors believe that for full data-sharing this figure is nice to be kept.

• The last chapter of the Abstract contains a duplication in line 36. Similarly, the same statement is given in line 123 and on page 11, line 12.

Response: The authors thank the reviewer for this comment; we have now adapted the manuscript accordingly to remove the repetitions.

• The Discussion is for the most part a repetition of Results. I think it would have been important to discuss the at least the APP gene polymorphisms in more detail including the description and citation of the first report of reduced APP activity in patients with ACEi-angioedema (pubmed/12086766).

Response: The authors thank the reviewer for this constructive comment; we have now revised the whole discussion section accordingly. A section on APP gene has been added to the discussion, including the study by Adam et al, Lancet, 2002. We believe this has improved the manuscript. 

• On the other hand, the part of the discussion about the results of Hubers et al. is not correct as an increased concentration of bradykinin isn’t necessarily caused by an impairment of bradykinin degradation (please refer to HAE). These authors based their conclusion on two further results, i.e. no change of the concentrations of the bradykinin degradation fragment B1-5 and the HMWK degradation product generated by kallikrein.

Response: The author agree that there is no definite conclusion that the increased level of bradykinin is due to the decreased degradation rates and it may also be caused by increased production rate. The manuscript has been amended accordingly.

• Reviewer #2: This article represents the first systematic review on genetic markers in ACEi-AE and further discuss the methodology used. It properly addresses relevant points in the evaluation of the selected articles, the weakness of each studies and limitations imposed.

In the abstract, I expected to see a summary of the reliable genetic markers associated to ACEi-AE. But the only conclusion made was about the quality of the studies. I suggest including some relevant data regarding genetic susceptibility to ACEi-AE, as implied in the title and stated in the final conclusions.

Response: The authors agree with this comment and changes have been made to the abstract. However no firm conclusions can be drawn, so how reliable the markers are is also discussed.

• Anecdotal evidences suggest that patients which present angioedema months after the withdrawn of ACEi cannot be classified as ACEi-AE but belong to another angioedema type. Although rare, a few reports support this hypothesis, describing FXII-HAE individuals that have been asymptomatic during their whole lives and only presented angioedema at advanced age when taking ACEis.

Response: This has been included in the discussion and is a very valid point. This interesting case has been included as reference: Veronez CL, Serpa FS, Pesquero JB. A rare mutation in the F12 gene in a patient with ACE inhibitor-induced angioedema. Ann Allergy Asthma Immunol 2017; 118(6): 743-5. 

• The use of Q-genie is interesting and strength the reliability of the selected articles. My only concern is that the focus of the article seems to be the methodology used and not the results of the analyzed studies. I expected a deeper discussion on the polymorphisms studied and the positive association found between them and ACEi-AE.

Response: The authors agree and have now added a new part to the discussion where the polymorphisms are discussed and their positive association found between them and ACEi-AE.

• Although Moholisa`s study find a significant association of +9/-9 genotype with both ACEi-cough and ACEi-AE, this association was not significant for the genotype -9/-9. Another point not highlighted is the fact that the control group was not in Hardy-Weinberg Equilibrium. The fact that Bas et al (2010) study did not found a significative association between this polymorphism and ACEi-AE also weakens its hypothetical influence in ACEi-AE. Another point to emphasize is the fact that –9/+9 was also never significantly associated with worst prognosis in HAE, where bradykinin role is quite clear. Should you surely conclude the -9/+9 polymorphism to be associated to ACEi-AE?

Response: The authors agree and a section on this has been added in the discussion.

• I disagree that the word-count limit in scientific journal represents a limitation in properly reporting methods, as mentioned in lines 117-118. Currently, even articles published as short communications or brie reports have almost unlimited space in supplementary material.

Response: The authors agree, we have now adapted the manuscript accordingly.

• As well pointed out by the authors, ACEi-AE in quite rare, hampering the power of all the analyzed studies. Besides the methodological recommendations for better organization to improve the quality of studies, the creation of consortium among expert groups on ACEi-AE could represent a big step into a larger and conclusive study on ACEi-AE susceptibility.

Response: This is very true, thanks for pointing this out, it has been implemented. Some of the authors are already involved in the PREDICTION-ADR consortium and genetic samples from our original study (unpublished data) have been shipped to Germany for inclusion in another larger study of ACEi-AE genetic markers. So things are improving, we believe.

• Minor

Line 43: Replace “ACEis inhibits the…” by ACEi inhibit the…”. 

Line 22: The reference 3, shown in the end of the sentence, should be withdrawn. Bas et al (2010) didn`t found association between -9/+9 polymorphism and ACEi-AE. 

In the Discussion (line 46), the authors say “The polymorphism of ACE I/D were significantly associated with AEi-AE. I think you wanted to say, “were not significantly associated”, right? 

Response: Yes, we have now inserted a “not” in the sentence.

---

## [Decision Letter · Decision Letter 1]

24 Oct 2019

Genetic susceptibility to angiotensin-converting enzyme-inhibitor induced angioedema: a systematic review and evaluation of methodological approaches

PONE-D-19-22350R1

Dear Dr. Rasmussen,

We are pleased to inform you that your manuscript has been judged scientifically suitable for publication and will be formally accepted for publication once it complies with all outstanding technical requirements.

With kind regards,

Michael Bader

Academic Editor

PLOS ONE

Additional Editor Comments (optional):

Reviewers' comments:

Reviewer's Responses to Questions

**Comments to the Author**

1. If the authors have adequately addressed your comments raised in a previous round of review and you feel that this manuscript is now acceptable for publication, you may indicate that here to bypass the “Comments to the Author” section, enter your conflict of interest statement in the “Confidential to Editor” section, and submit your "Accept" recommendation.

Reviewer #1: (No Response)

Reviewer #2: All comments have been addressed

2. Is the manuscript technically sound, and do the data support the conclusions?

Reviewer #1: Partly

Reviewer #2: Yes

3. Has the statistical analysis been performed appropriately and rigorously? 

Reviewer #1: N/A

Reviewer #2: N/A

4. Have the authors made all data underlying the findings in their manuscript fully available?

Reviewer #1: Yes

Reviewer #2: Yes

5. Is the manuscript presented in an intelligible fashion and written in standard English?

Reviewer #1: Yes

Reviewer #2: Yes

6. Review Comments to the Author

Reviewer #1: I have no further comments, although I still think that is does provide only minor additional information.

Reviewer #2: (No Response)

7. PLOS authors have the option to publish the peer review history of their article (what does this mean?). If published, this will include your full peer review and any attached files.

Reviewer #1: No

Reviewer #2: No

---

## [Editor Report · Acceptance letter]

30 Oct 2019

PONE-D-19-22350R1 

Genetic susceptibility to angiotensin-converting enzyme-inhibitor induced angioedema: a systematic review and evaluation of methodological approaches 

Dear Dr. Rasmussen:

I am pleased to inform you that your manuscript has been deemed suitable for publication in PLOS ONE. Congratulations! Your manuscript is now with our production department. 

With kind regards,

on behalf of

Prof. Michael Bader 

Academic Editor

PLOS ONE